# Automatic Management and Monitoring of Bridge Lifting: A Method of Changing Engineering in Real-Time

**DOI:** 10.3390/s19235293

**Published:** 2019-12-01

**Authors:** Yao Min Fang, Tien Yin Chou, Thanh Van Hoang, Bing Jean Lee

**Affiliations:** 1Geographic Information Systems Research Center, Feng Chia University, Taichung 40724, Taiwan; frankfang@gis.tw (Y.M.F.); jimmy@gis.tw (T.Y.C.); 2Department of Civil Engineering, College of Construction and Development, Feng Chia University, Taichung 40724, Taiwan; bjlee@fcu.edu.tw

**Keywords:** bridge dynamics, lifting method, structural health monitoring (SHM)

## Abstract

In recent years, owing to the increase of extreme climate events due to global climate change, the foundational erosion of old bridges has become increasingly serious. When typhoons have approached, bridge foundations have been broken due to the insufficient bearing capacity of the bridge column. The bridge bottoming method involves rebuilding the lower structure while keeping the bridge surface open, and transferring the load of the bridge temporarily to the temporary support frame to remove the bridge base or damaged part with insufficient strength. This is followed by replacing the removed bridge base with a new bridge foundation that meets the requirements of flood and earthquake resistance. Meanwhile, monitoring plans should be coordinated during construction using the bottoming method to ensure the safety of the bridge. In the case of this study, the No. 3 line Wuxi Bridge had a maximum bridge age of 40 years, where the maximum exposed length of the foundation was up to 7.5 m, resulting in insufficient flood and earthquake resistance. Consequently, a reconstruction plan was carried out on this bridge. This study took the reconstruction of Wuxi Bridge as the object and established a finite element model using the SAP 2000 computer software based on the secondary reconstruction design of the Wuxi Bridge. The domestic bridge design specification was used as the basis for the static and dynamic analyses of the Wuxi Bridge model. As a result of the analysis, the management value of the monitoring instrument during construction was determined. The calculated management values were compared with the monitoring data during the construction period to determine the rationality of the management values and to explore changes in the behavior of the old bridges and temporary support bridges.

## 1. Introduction

Structural Health Monitoring System (SHMS) bridge construction monitoring systems have started to be applied and developed throughout the world in recent years. Most of the major bridge projects around the world are installed with different monitoring systems to continuously monitor and collect data (physical quantities) during the operation and exploitation of bridges. The US [1,2], Japan [3,4,5] China [6,7,8,9], and Europe [10,11,12] are places where monitoring systems are widely and effectively applied.

Monitoring systems are designed specifically for each bridge construction based on the structural characteristics of each project and the financial situation required by the investor. The bridge monitoring system must be highly durable with a high level of accuracy throughout the operation time. The cost of the entire monitoring system is not large compared with the total cost of bridge construction, accounting for 0.3% to 1.5% of the total value of the bridge construction investment, depending on the complexity of the system monitoring. The management and operation of the monitoring system is also not a large cost compared with the total cost of maintaining the works, but the monitoring system requires a high level of personnel as well as management experience.

The biggest advantage of the bridge construction monitoring system is the continuous monitoring of the structure’s activities and changes allowing the more effective and safe operation and exploitation of the works. Based on the analysis and evaluation of monitoring data [13], we can make the right decisions for the maintenance of the bridge works. The monitoring results allow us to assess the correctness of the hypotheses given in the design and construction process. Regular monitoring of the work allows us to control the operation of the building under the influence of different load combinations, helping experts identify the aging processes of structures and ensuring the implementation of measures to prevent and improve the service life of bridges [14,15,16].

The bridge construction monitoring system is a complex system that combines many components from construction structural monitoring, meteorological monitoring, and image monitoring to geographic monitoring. Monitoring plays an important role in the processes of construction, construction, and operation. It allows hypothetical conditions to be set during design and can affect the construction cost of the project. Therefore, the efficient and safe implementation and application of modern methods and advanced techniques to monitor bridge works for the construction, research, and management of works are essential and urgent.

The discovery of the spoilage of the bridge works before there are specific signs is one of the issues that is attracting the attention of many bridge researchers and engineers. Along with the development of science and technology, calculation software and simulations have been developed, but the received results of theoretical calculations are only relatively close to the actual behavior of the project. In the design process, setting up the hypotheses to simplify the calculation model leads to the issue that analysis and calculation cannot reflect the status of operation and behavior of the project under normal operating conditions, as well as in terms of exploitation. In addition, there is a big difference between simulation in design and reality. In order to determine this difference, one of the measures with significant potential to assess the process of the work and exploitation of bridge works is to install monitoring devices at reasonable locations related to the project’s demand to continuously measure the parameters affecting construction such as external forces (wind, earthquake, vehicle load) and behaviors of bridge construction (oscillation, displacement, stress) [17,18,19,20]. The data collected will be the basis for checking the process of the analysis and calculation of works. Besides, the bridge monitoring system is also used for other purposes such as maintenance, construction, and traffic management on the bridge.

Because of their particularly important role and costly construction, large-span bridges often have to be checked regularly to ensure that they are safe [21]. This testing process usually has high time and cost requirements, but sometimes it is still impossible to control all the risks of damage to the bridge. The construction and development of an automatic bridge monitoring system is something that developed countries such as the US [22], Europe [23], South Korea [24], and China [13] have been doing to reduce the costs of maintaining bridges as well as ensuring traffic safety for these works.

In the United States, in the mid-to-late 1980s, monitoring devices were installed on a number of large bridges. For example, more than 500 sensors were installed on the 440 m Sunshine Skyway bridge in the United States to monitor the state of the bridge [25]. Monitoring research on the condition of bridges originated in the Europe and America in the mid-to-late 1960s–1980s [26]. The United Kingdom also deployed sensors on the Foyle bridge, which is a three-span high-rise steel box girder bridge, with a total length of 522 m, to monitor the bridge in the operational stage. The monitored section responds to the vibration, deflection, and strain of the main beam under the action of the wind load, and the temperature of the environment and the temperature of the structure are also monitored. This system was one of the earliest installed and relatively complete monitoring systems, which achieved the purpose of real-time monitoring, real-time analysis, and data network sharing [27].

Hong Kong’s Tsing Ma Bridge had an anemometer and an accelerometer installed on the bridge to establish a monitoring system regarding the wind and structure. The wind field characteristics and damping ratio when the typhoon actually attacked were matched with the parameters obtained by the wind tunnel test [28]. The Chinese Tiger Gate Bridge and the Jiangyin Yangtze River Bridge are also equipped with corresponding monitoring systems. The Humen Bridge monitoring system is composed of strain gauges, acceleration sensors, temperature sensors, displacement sensors, GPS systems, etc. Based on the construction monitoring and the test system after the completion of the bridge body, the monitoring of the bridge after the bridge was opened to traffic has produced many positive effects to ensure the safe operation of the Humen Bridge.

In view of the lack of seismic capacity of bridges caused by the bare foundation of the bridge, structural reinforcement measures are applied for bridge foundation reinforcement protection, bridge demolition, and reconstruction. Alternatively, local reinforcement bridge foundation methods, such as concrete piers and supports, can also be applied. Amongst others, the bottom expansions are all based on the bridge structure and river characteristics, and each has its own applicability.

Wuxi Bridge is an important bridge connecting the Wufeng District of Taichung City with Caotun Town of Nantou County. It was rebuilt after the 921 (21 September) earthquake in the Republic of China. In recent years, the riverbed has been continuously washed away by floods, and the bridges between P9 and P13 have been cavitated.

The foundation is severely exposed; therefore, the vertical bearing capacity and lateral bearing capacity of the old foundation have been significantly reduced. Piles have been used to protect and reinforce the construction method several times, but the bridge is still affected by the continuous decline of the riverbed and the lateral erosion of the river channel. Owing to safety concerns, it is listed as a damaged bridge in the province. To improve the overall seismic capacity and flood resistance of the bridge, the superstructure of the project was rebuilt by the Republic of China in 1991, and the structure is still new and good. Therefore, under the conditions of using the existing superstructure and maintaining the original traffic conditions, the partial reconstruction of the bottom method was applied.

In recent years, bridges in Taiwan have been washed away. In the Republic of China in 1985, the typhoon He Bo caused severe collapses and sloping damages to the main bridges in the west. This included strong bridges that were as high as 6.3 to 9.3 m, and the southern Ligang Bridge was also damaged by the slope of the bridge. This resulted in traffic disruption [29].

In 1990, the Xizhou Bridge on the first line was affected by typhoon Taozhi in the Republic of China. As a result, the depth of the riverbed at the bridge site of the Xizhou Bridge was reduced by more than 5 m, and the bridge foundation in the deep trough area was seriously exposed by up to 10 m, which was unsafe. To maintain bridge safety and to maintain unobstructed bridge traffic, the first bridge replacement method was carried out in Taiwan. Under the conditions of maintaining the structure and traffic of the bridge, the bridge load was temporarily transferred from the original pier to a temporary supporting steel frame, and the bareness was addressed. The damaged bridge foundation was replaced with the new bridge foundation, which met the requirements of flood resistance and seismic design. The difference between the bridge deck before and after construction was extremely small, and the shape and function of the original bridge could be maintained to effectively resolve the bridge erosion. Considering the problem of transportation disruptions and the advantages of saving engineering costs and construction period, and considering environmental protection and energy saving, environmental beauty, and ecological maintenance, this approach provides a good solution to the problem of bridge foundation exposure caused by domestic erosion through water damage [30].

Jingzhou Xin et al. [31] proposed the Kalman–ARIMA–GARCH model (autoregressive integrated moving average model (ARIMA), and generalized autoregressive conditional heteroskedasticity (GARCH)) to predict the deformation of bridge structure based on GNSS (Global Navigation Satellite System). Chiara Bedon [32] explored the use of MEMS (Micro Electro-Mechanical Systems) Accelerometers for prototyping and validation for structure health monitoring in Italy. John Reilly et al. [33] gave further evidence on how identifying temperature can affect structure health monitoring. Zengshun Chen et al. [30] have combined three methods—the peak-picking method, the random decrement technique, and the frequency domain decomposition—for a smart structure health bridge monitoring system. Olga Thalla el al. [34] performed an experiment in which they analyzed data to detect damage and combine the monitoring of wind data to explain the sequence of events.

The newest point of this study is that the research integrates the monitoring system (in real-time) using the computer software tool SAP200 to carry out a safety analysis of the bridge structure in the case study, namely that of Wuxi bridge.

## 2. Introduction to the Method

### 2.1. Method of Changing the Bottom of the Wuxi Bridge on Provincial Highway 3

In general, bridge reconstruction must be completed with full bridge closure, resulting in traffic disruption and inconvenience to passers-by. In this paper, the case study focused on the “Zhoudaotai 3 Line Wuxi Bridge”, which is an important contact bridge between the Caotun area and the Taichung area, where the traffic volume is very large. Therefore, the bridge replacement method was used to ensure the maintenance of the original traffic during the bridge reconstruction period. 

### 2.2. Introduction of Wuxi Bridge’s Bottoming Method

During the reconstruction, six temporary support piles were erected around the foundation of the substructure (bridge foundation, bridge column, and cap beam), and a temporary cap beam structure (steel structure) was set up on the pile column as a pier (column). The temporary support system during the demolition and rebuilding period is shown in Figure 1. This was added to the girders of the jacking jack, and a temporary support stiffening plate for the girders was added to increase the stiffness of the steel beams supported by the temporary cap beam steel frame. To ensure the safety of the bridge structure during jacking, a monitoring system was installed near the lifting device. The monitoring equipment includes an electronic tilt meter, electronic sinker, etc. to monitor the safety of the bridge.

The jack is shown in Figure 2, and it was placed on the steel frame of the temporary cap beam. When the jacking was carried out, the section was lifted up, and after a cumulative increase of 50 tons (each jack load), the structure was stabilized after pausing for half an hour. At the same time, the measurements of each jack and the displacement of the beam were recorded. After that, the jacking was carried out, so that when there is an obvious separation of the old pier support, the lifting was stopped, and the steel spacers and anti-seismic steel structure were fixed. At this time, the weight of the bridge was transferred to the temporary supporting steel structure. 

After the load was transposed, the old bridge pier and foundation could be removed. After the new foundation, pier, and cap beam were completed, the load was transferred to the new pier and the six temporary support piles could be removed. The temporary support of the steel frame was raised to complete the reconstruction work.

### 2.3. Structure Analysis (Computer Software Tools SAP2000) 

The SAP2000 computer software (COMPUTERS & STRUCTURES, INC., Structural and Earthquake Engineering Software, Walnut Creek, CA, USA) is powerful full-window interface structure analysis software and is capable of the establishment of basic three-dimensional geometric shapes, the cross-sectional properties of rods and thin shell elements, the mechanical properties of reinforced concrete, steel structures, nonlinear elements, or the new definition of material properties. Even static analysis, dynamic analysis, response spectrum analysis, and diachronic analysis can be performed using SAP2000 computer software, and the analysis results can be displayed graphically or in a standardized text format for subsequent processing work.

This study used the finite element structure analysis in SAP2000 computer software to carry out the safety analysis of the Wuxi Bridge structure. The establishment of the FE (Finite Element) bridge model algorithm can be divided into the “whole bridge system” and the “single partial system”, which is the same as the vibration unit concept of the current bridge design code. The Finite Element “whole bridge system” model is suitable when the bridge type is geometrically irregular, such as for a cable bridge, which has a horizontal multi-channel expansion joint, and when the bridge is located in a soft soil. The “single partial system” is suitable for quantifying the strength and stiffness capacity of a single frame, such as piers, and involves vertical and horizontal analysis. The longitudinal axis should consider the adjacent-span effect and depends on the bridge length. The transverse axis also considers the adjacent-span effect. The upper structure can be regarded as a rigid member. The overall bridge analysis model must accurately describe the dimensions of all components, such as structural elements, thin-shell elements, springs, bearings and expansion joints, and other elements. The material properties comprise the behavior of the intact reaction structure. The section of the upper structure of the bridge is calculated to determine its section properties. According to the results of an on-site microseismic experiment of reinforced concrete bridges, the upper structure is pre-force concrete. The torsional stiffness parameter is calculated in 200% of the full section. The flexural rigidity of the horizontal axis parameter is calculated in 120% to 140% of the full section. The flexural rigidity of the vertical axis parameter is calculated in 100% to 120% of the full section. The cross-section of the rigid element is magnified 1000 times by the cross-section of the beam-column element. The design of the bridge reconstruction project after the 921 Jiji Earthquake and the completion map of the bridge reconstruction project were used to construct the Wuxi Bridge. The structural analysis model was used to perform microseismic tests on each span of the bridge (P9~P15) to modify the model constructed by the SAP2000 software, in order to make the model fit closer to the actual condition of the bridge.

### 2.4. Application of Monitoring System

Bridges are an indispensable transportation lifeline for all nations. However, given old bridge structures and the impacts of earthquakes and typhoon disasters over the years, the strength of bridge structures has decreased, and their life expectancy has been reduced. Thus, the safety of passers-by has also been jeopardized. There is still an unknown risk when a bridge is damaged. For example, on 28 August 1989, the Gaoping Bridge—an important transportation bridge in the southern part of the southern line—was broken and the bridge deck was slanted, causing 16 large and small vehicles to fall into the stream, resulting in 22 deaths and injuries. On 14 September 1997, the Hehou Bridge—an important transportation bridge on the 13th line of Central Taiwan Station—was also severely eroded due to the bridge foundation. The bridge pier was weak and the bridge was broken, causing six people to fall below.

Owing to the influence of weather and human factors, the lack of a monitoring system means that damages cannot be detected, thereby reducing the functionality of bridges. The level of bridge safety is unknown, which often causes serious damage to traffic and passers-by. Therefore, a monitoring system is required to monitor the condition of the structure and to evaluate the safety of the structure. Field verification of design theory has been widely used to facilitate the understanding of the structural characteristics of the bridge and for the improvement of the design technology.

## 3. Local Experiment

### 3.1. Experimental Planning

Since the research project aimed to change the bottom of the bridge while it was being used, when the actual load was applied to the bridge, the measurement results of various maintenance management projects would also contain dynamic components, and most of these dynamic values were monitored and obtained. In addition, the results from the vibration monitoring during reconstruction could be used to determine and review the software model that was in use, or to set the initial characteristics of each monitoring project; that is, to apply these monitoring results for evaluation or research. The local experiments at the research project comprised the microseismic measurement experiment and the strain measurement experiment. The information obtained from these experiments further assists in the accuracy of the computer software model and in the management values.

#### 3.1.1. Microseismic Measurement Experiment

Among all the dynamic tests, the only requirement is to disregard or input any disturbance, and the test process is very convenient, as it does not need long-term measurement. The analysis method can quickly and accurately reflect the bridge’s state. The microseismic measurement test can be run during the construction process (Figure 3 and Figure 4), where the measurement results and data are used to correct the numerical model of the bridge constructed by the computer software, including the vibration, frequency, and vibration mode. We conducted a microseismic measurement experiment which included an environment test and a test of bridges. The environment ambient vibration test was carried out first before the bridge was constructed and was performed after each ambient vibration test. Its goal was to verify the bridge frequency and identify the bridge vibration mode and to modify the bridge model accordingly.

#### 3.1.2. Strain Measurement Experiment

As part of the jacking and bottoming method, it was planned to place 24 jacks on each temporary cap beam, each of which was 208 cm apart. This was followed by the attachment of 24 strain gauges (see Figure 5, Figure 6 and Figure 7) next to each jack position before jacking. When jacking up, you can obtain the force corresponding to the jack and the strain caused on the support frame. This helped in analyzing the distribution of forces transmitted to the support frame during the jacking process and determining whether it caused dangerous conditions, such as the tilting or sinking of the bridge. The force of the steel beam was temporarily supported before and after the jacking process, and the reading value of the strain gauge changed significantly (Figure 8).

#### 3.1.3. Measured Value

The microseismic measurement was divided into the measured values of the original bridge and the bridge after jacking. The measuring points were all in the middle section of each span. The sampling frequency was 400 HZ, and the data were divided into three axial directions. They were X (north–south direction), Y (east–west direction), and Z (vertical direction). After the FFT (Fast Fourier Transform) was obtained from the time domain data obtained from the experiment, the natural vibration frequency of the bridge was obtained (see Figure 9, Figure 10 and Figure 11, Table 1). The lifting of the bridge will change the bridge vibration modes and frequencies. These field experiment data were used to modify the vibration frequency of the SAP2000 bridge structure model, meaning that the model established by the SAP2000 software was more in line with the current analyzed situation. To identify the frequencies of the bridge, the signals collected in the time domain were transformed to the frequency domain by using the Fast Fourier Transform (FFT), as shown typically in Figure 9, Figure 10 and Figure 11. Many peaks appeared in the Fourier spectrum diagram due to the disturbance and noise in the field.

### 3.2. Instrument Planning

In the static measurement part, the instruments used in this project included a subsidence meter and an inclinometer, and 42 structural subsidence points were set, respectively. Seven piers from P9~P15 were planned to be installed at the subsidence point, each with a temporary cap on the beam steel frame. There were six subsidence points, and another 28 subsidence points were set on the bridge deck. The planned installation position was on the bridge deck above the P9~P15 pier. The bridge deck above each pier included two groups of subsidence points. In order to ensure the safety of the upper structure of the bridge during the jacking process of the superstructure, 28 connected tubular subsidence gauges were installed, and the planned installation position was on the outer side of the deck above the P9~P15 pier (outside of the New Jersey-style guardrail). The bridge spanned the middle.

The inclinometer was installed on the pier. Except for each temporary cap beam steel frame, each device had four inclinometers (28 piers total), and 14 automatic electronic inclinometers were planned, which were installed on the P9~P15 piers above the steel beam (two per pier). The purpose of the main measurement was to measure the duration of the upper structure during the jacking process of the upper structure, and the installation position is shown in Figure 12.

After the completion of the monitoring system architecture, the integration of each sensing device interface was carried out, and the remote and complete monitoring system operation interface was provided to facilitate the follow-up maintenance and data analysis by staff.

### 3.3. Measurement Method and Instrument Description

#### 3.3.1. Subsidence Meter

1. Monitoring purposes:

The working principle was similar to that of the displacement meter; however, the displacement measured by the subsidence meter was mostly the absolute displacement amount; that is, a reference point needs to be set at the initial stage of monitoring, and this reference point is used as the linear coordinate origin in future monitoring to determine the subsidence amount. The installation location was the pier foundation or abutment position to monitor the possible subsidence of the overall bridge. The piers planned for installation in this research project’s subsidence plan were P9~P15.

2. Connected tubular subsidence meter (automatic monitoring):

At present, the most common approach is the analysis of the relative subsidence of the connected pipe principle; that is, installing a small water tank and a water pressure gauge at approximately the same height as each measured point, and then connecting the water pipes to a liquid communication system at one of the measuring points. When subsidence occurs, the water level in the sink increases and the water pressure rises, which translates into a relative subsidence. The temperature effect and the pressure head loss caused by the friction of the pipe wall when the pipeline is too long should also be taken into consideration. The liquid mixed with glycerin and water at a ratio of 1:1 can be used to reduce the influence of the temperature difference. This equipment is installed below the outside of the bridge guardrail. The main instrument components of the monitoring system are the connected tubular subsidence meter and the reference water tank. The measuring range was ±50 mm or more, the precision was less than 1% of the maximum measuring range (i.e., less than 1%), and the automatic recording was maintained for 24 h.

3. Bridge deck test sink (manual monitoring):

To avoid damage to the main structure of the bridge, this monitoring system used round-head galvanized steel nails (about 20 mm long) as the sinking base point, and the installation position was according to the requirements of the design specifications.

4. Structure sinking point (manual monitoring):

This item was monitored on the temporary RC (Reinforced Concrete Square Pile) pile, and the fixing bolts were locked, and the standard method of painting was used as the subsidence base point. The installation position was installed according to the requirements of the design specifications.

The schematic diagram of the instrument installation is shown in Figure 13, and the instrument configuration is shown in Figure 14.

#### 3.3.2. Tilt Meter

1. Monitoring purposes:

The inclinometer is mainly used to monitor the lateral inclination of the structure, and the measurement direction is divided into one direction and two directions. Owing to the long-term exposure of the pier foundation, the focus of the monitoring is on whether the bridge has a certain degree of inclination, and thus a change in bearing capacity. In this study, an electronic inclinometer was placed in the middle of the bridge cap beam to evaluate the tilting condition of the bridge. The dip angle value of each bridge span was measured using a real-time monitoring system, and the safety warning was evaluated.

2. Electronic inclinometer (automatic monitoring):

This is generally installed on the column of the structure around the foundation excavation to measure the degree of tilting of the surrounding surface caused by excavation, pumping, or other factors, resulting in the inclination of the structure. The main principle is that when the structure is tilted, the tilt meter installed thereon is also inclined, and the voltage value generated by the structure is transmitted to the recorder and then converted into a tilt angle. The device is installed under the outside of the bridge guardrail, the measuring range is ±5 degrees or more, and the automatic recording is maintained for 24 h.

3. Structure inclinometer (manual monitoring):

In principle, the inclinometer is set at the designated location to measure the inclination of each direction of the structure. The components mainly include the inclinometer body, the protective cover, the support frame (durability and rust prevention), and all fixed components. The inclinometer sensor (SENSOR) and the indicator (INDICATOR) are combined with the measurement range of ±5 degrees or more, and the accuracy is 2% or less of the maximum measurement range (i.e., less than 2%).

The schematic diagram of the instrument installation is shown in Figure 15, and the instrument configuration diagram is shown in Figure 16.

## 4. Theoretical Analysis

### 4.1. Static Analysis Process

We used a design diagram of the bridge to build a bridge model by Sap2000. According to the vibration mode shapes and vibration frequencies of the in-field test results, we modified the bridge model. We used static management to analyze the management values of the positioning shift meter and the tilt meter. According to the bridge design specifications and the seismic design specifications, static forces such as dead load, live load, wind power, and seismic force were input into the SAP2000 computer software simulation model for static analysis.

In addition, according to the strength design method of the Highway Bridge Design Code, the relevant data were input into the SAP2000 computer software, and each load combination under the force combination of the bridge model was simulated to carry out structural analysis.

### 4.2. Power Analysis Process

The dynamic analysis system used the SAP2000 computer software to calculate the seismic acceleration time-of-day analysis to simulate the monitoring and management values affected by the earthquake during the bridge jacking. Since the fog peak and the grasshopper belong to the earthquake zone A, the designed ground acceleration was 0.33 g. The 921 Jiji earthquake near the Wuxiqiao seismic station was TCU065 (Yufeng Guoxiao station). The acceleration data was linearly amplified to 0.33 g in the horizontal direction and linearly amplified to 0.22 g, as in [6], in the vertical direction. The three directions were simultaneously input into the SAP2000 computer software for analysis.

We found the position of the sinker and the inclinometer in the SAP2000 computer software model, and after analysis, the displacement of the bridge under the earthquake could be obtained.

### 4.3. Determination of Management Values

Based on the established SAP2000 computer software model, the analysis model for monitoring the management value research was constructed according to the Highway Bridge Design Code promulgated by the Ministry of Communications in January 1990 [35] and the Highway Design Code for Highway Bridges promulgated in April 1989. Based on the static and dynamic analysis results of the computer software model, the monitoring and management values of each monitoring instrument were determined, and the preliminary monitoring management values were compared to the collected monitoring data to ascertain the reasonableness of the monitoring management values.

Based on the static and dynamic analysis results of the deformation, the monitoring, and the management value of each monitoring instrument were determined. According to the bridge design code, the bridge will bear the load data and combination. We consider that the maximum value of deformation is the safety value, the maximum value divided by 0.9 is the alert value, and the maximum value divided by 0.75 is the action value. The process of setting the management values (Figure 17) showed the strength reduction system, with 0.9 and 0.75 as the strength reduction system values in the reinforced concrete strength design method [6].

Comparing the management values and the actual measurement results, it could be confirmed from the figure that the measurement results did not exceed the management values set. Thus, the management values are reasonable and can be used as a warning reference.

### 4.4. Bridge Structure Simulation and Vibration (Mode) State Analysis Results

According to the original design drawing and reconstruction plan of the Wuxi Bridge on the provincial line 3, the bridge model was built using the SAP2000 software. When the bridge was simulated by finite elements, the weight and stiffness of the bridge was different from that of the real bridge. In order to ensure that the model could represent the real bridge, the data and parameters obtained from the local experiment were required to be matched, including the vibration frequency and vibration mode, etc., and the structural model was modified to make the vibration frequency and vibration mode of the model more in line with the current state of the bridge. This would help to more accurately analyze the problems of the mechanics, safety, and lack of workmanship of the bridge. Before, during, and after the bridge lifting method, the bridge model must be modified separately, and the original bridge column should be replaced with a temporary support frame and a new pier, along with the model of the bridge simulation at different periods (Figure 18, Figure 19, Figure 20, Figure 21, Figure 22, Figure 23, Figure 24 and Figure 25). Figure 18 and Figure 19 are model 1 and model 2 of the original bridge. These use northbound and southbound *Z*-axes. After the bridge jacking and lift, the vibration mode of the bridge will be changed. The direction of the different vibrations can be seen in Figure 20 and Figure 21. Different bridge pillars will also cause different modes, as shown in Figure 22, Figure 23, Figure 24 and Figure 25.

## 5. Monitoring System

### System Architecture

The software environment of the project monitoring system was as follows: web platform: IIS (Microsoft Internet Information Services) 6.0, programming language: ASP.NET (Active Server Page of Microsoft), database: SQL Server Express. The hardware environment was ADSL (Asymmetric Digital Subscriber Line), server, hub. The software architecture of the project monitoring system is shown in Figure 26.

The functional planning of the project monitoring system was divided into the following nine functions:Data conversion and storage function: The system automatically calls the sensor to obtain the sensing data according to the set value and writes the corresponding data table according to the different types of sensors, where the written information includes the sensor number, data time, sensing data, and data status.System main interface: This is divided into the login interface and system home page, as shown in Figure 27 and Figure 28.Authority management and maintenance function module:(1)Add account: This allows the application interface on the homepage to add the user’s account, including the account number, password, unit, name, mobile phone, and other information.(2)Account maintenance: Users can modify the data in their own account, but the account cannot be modified, and the modification level must be modified by the system administrator.(3)Privilege Control: For the inquiry account interface, this is used for the account verification function and the account level setting function, which can set the level of the account and also delete the account data.(4)Login system function: This provides the account password login function on the homepage, compares the account passwords sent to enter the system, and provides the “forgot password” function in which the user enters their account number and email address. If the match is met, the system will send the password to the email address and provide system functions according to the authority set.Monitoring point management function(1)Function entry: Monitoring point management.(2)Query interface: This provides the bridge pier number and instrument category query conditions, and it also provides the new, modify, delete, print (Excel) functions, where the system administrators and data maintainers can edit and delete data, and the system automatically records and edits personnel. When deleting the data, the system will display a confirmation window for the user to confirm, and the device can be deleted if are is no data in the instrument. After confirming the deletion, the “data deleted” dialog box will be displayed, as shown in Figure 29.(3)New monitoring point data: This is used to distinguish between automatic measuring points and manual measuring points, as well as adding monitoring point data, including the bridge pier number, measuring point number, measuring point type, sensor number, and other data. When the new measurement point number is added, the existing number will be automatically avoided, and the recommendation number will be automatically displayed. The suggested number is modifiable. After the confirmation number is sent, the system will automatically generate the identification number (SID, System Identification) and conduct an automatic check. Whether the measuring point SID is repeated, if it is repeated, it returns to the editing screen and is marked with a red word after the repeated data. If it is an automatic monitoring point, the system checks whether the signal channel was repeated. If it was repeated, the system will return to the editing screen and it will be marked with a red word after the repeated data. When the data are written into the database after it has been checked, the dialog box “The data has been added” will be displayed. If it is a manual monitoring point, there is no need to fill in the signal channel, as shown in Figure 30.(4)Monitoring point data maintenance: After clicking the measuring point number in the query list, the editing interface icon will be opened, which is the same as the new interface graphic structure; however, the pier number, measuring point category, and measuring point number are not modifiable. When the data are written to the database through the check, the dialog box “Data has been modified” will be displayed. This function also provides the function of deleting the monitoring point data, although the monitoring point cannot be deleted if the sensor data has been received or the observation data has been reported.Manually report the monitoring data maintenance function module:(1)Function entry: Manual monitoring and maintenance.(2)Data inquiry: After clicking the bridge pier number, the system will automatically enter the manual measurement point number. In the time interval, a small calendar will be provided for selection. If the time is not selected, it will represent the whole day. When a date is specified, it will only check the data for that day. In addition, when the “normal value” is checked, the normal value data will be displayed; if the “exceeding the warning value” is checked, the data exceeding the warning value will be displayed; and if the “exceeding the action value” is selected, the data exceeding the action value will be displayed. For all three, the user selects at least one option to check. When the user clicks “Query”, a list of information will be displayed. When they click the new icon, the “Add” window will open. The query list is printable. After selecting the measurement point number, the system will automatically display it, providing a description of the measurement point.(3)New monitoring data: When using the supervision unit and the bridge name number, these will be automatically displayed, along with the other pier number selections, measuring point selection, observation time, observation data, reporting personnel, and reporting time. When selecting the measuring point number, the system will automatically display the description information of the measurement point.(4)Monitoring data maintenance module: When clicking the editing icon of the measuring point number in the query list, the editing interface will be opened, which is basically the same as the new interface, only that the time and value are allowed to be modified, whilst the pier number and the point number are not modifiable. After the data are written into the database, it will display the “Data has been modified” dialog box, and the system will automatically record the modified personnel. In addition, this function also provides the function of deleting single monitoring data. When deleting, the system will automatically record the personnel and time of deleting the data.Warning value management function module:(1)Function entry: Alert value management.(2)Warning value interface module: The supervisory unit and bridge name number will be displayed automatically when using the module, and it will provide the UI (User Interface) of the sinker, the tilt meter alarm value, and the action value input, as shown in Figure 31. The default mode of the alert mode is to check the send SMS option, and unchecking will not send the SMS. If the setting is enabled, the default value is ticked. When the alarm value is changed, the system will display the dialog box “The status of all monitoring data will be re-determined, whether the change is confirmed”. If the change is confirmed, the system will redetermine all the sensing data and redisplay the field data.Monitoring data query function module:(1)Function entry: Monitoring data query.(2)Seepage monitoring data query: This provides the conditions for the pier number, sensing method, measuring point number, time interval, and action warning. After selecting the conditions, the qualified information will be displayed. In addition, after clicking the bridge pier number, the system will automatically display the measurement point number. When the automatic or manual check is selected, the measurement point number will be automatically displayed. The default value is both full hooks. At least one option should be selected when using. In the time interval, a small calendar is available for selection. If the time is not selected, it means the full day. When a date is specified, it means only the day the information is checked. When the “normal value” is checked, the normal value data are displayed, and the warning is exceeded. “Value” displays the data exceeding the warning value. If the “Exceeded action value” is checked, the data exceeding the action value are displayed. The default value is fully selected. When using this, at least one option must be checked to query. When clicking “Query” to display the list of materials, while using the paging function, the output in Excel format data will be printed.(3)Tilt meter monitoring data query: This uses the monitoring data query like the sinker.Instant warning function module:(1)Warning resident program: For a resident in the operating system, the triggering time is as follows: after the sensing data of the module are captured regularly, the data input system function module is added to the data, and if the warning value is exceeded, the action is exceeded. The value is sent as a short message notification to the set alert object.Monitoring information display platform function module(1)Full-bridge monitoring information display module: This displays the full-bridge diagram of the construction scope of the provincial road “Taiwan 3 line 210k Wuxi Bridge Foundation Reconstruction Project”, including the direction and pier number. The piers included in the monitoring are marked with lights. When the bridge pier is normal, it is displayed with a green light, and when any sensor data of the pier exceed the warning value, a yellow light warning is displayed. When any sensor data of the pier exceed the action value, a red light warning is displayed, and all the sensor information is listed. The sensor information content includes the sensor name and the sensing data. The sensor data exceeding the warning value are represented by an orange indicator. The sensor data exceeding the action value are indicated in red, and the abnormal sensor data are in black. When the mouse cursor is moved to any pier, all the sensor information is displayed. Double-clicking the bridge icon opens the display module of the bridge monitoring information, as shown in Figure 32. In the illustration in Figure 32, the red warning of the P09 pier is due to P9 pier being used for the jacking project. After P09 pier is lifted, the reference value of the subsidence meter will be reset (before the bridge pier is replaced) to monitor the subsidence and inclination of the P09 pier after jacking.(2)Bridge pier monitoring information display module: This is displayed as a small picture with full bridge graphics, including direction and pier number. The bridge piers that are included in the monitoring are all displayed by lights. The bridge monitoring diagram can display three kinds of graphics: front view, side view, and plan view. The other sensors are displayed in different positions according to their types. When the sensor data bit is at a normal value, it is displayed as a green icon. When the sensor data exceed the warning value, they are displayed as a yellow icon. When the action value is exceeded, it is displayed as a red icon to inform the monitoring manager, and to inform the relevant personnel by SMS. The information of each sensor indicates that the monitor is still reminded by the color of the text. When the monitored value exceeds the warning value, the data displayed by the sensor are represented by orange text. When the monitored value exceeds the action value, the sensor displays the data in red text. When there is no abnormality in the monitored value, the data displayed by the sensor are indicated in black text. When the user double-clicks the full bridge icon, they will return to the full bridge monitoring information display module. When they double-click the sensor, the sensor monitoring information display module is turned on, as shown in Figure 33.

## 6. Conclusions and Recommendations

The traditional bridge detection method has the disadvantage of not having continuity, and the value of manual timing observation easily produces errors affecting the detection results. The automatic monitoring system can greatly reduce the monitoring errors and can continuously record the sensor data. It also has an instant warning function, which is a necessary trend going forward.At present, most of the bridge construction monitoring in this area still uses the traditional manual timing observation and measurement as the data analysis method. When a typhoon is heavy, or the river is flooded, the artificial measurement cannot be measured and observed. However, when the typhoon is rainy, this also means the bridge is dangerous. Therefore, in order to ensure that bridge monitoring is continuous and immediate, it is recommended to adopt a scientific automatic monitoring system to match modern trends.In order to understand the dynamic response of the bridge and the warnings of change, various types of monitoring sensors are installed on the bridge to obtain structural response and environmental data, and long-term analysis and evaluation work should be carried out to reduce the social cost caused by a disaster. Based on the concept of bridge monitoring center information management, and with the advancement of network technology and remote control and centralized management, integrating all bridge monitoring systems to achieve effective centralized management reduces the need for managers.This paper establishes the finite element computer model of the bridge and sets the management values for monitoring with reference to the bridge design specifications and design data, and it compares the management values with the actual monitoring data to determine whether the preliminary monitoring management value is reasonable. Therefore, it proves that the method of setting the management value is feasible.Regarding the impact of bridge use and vehicle driving and the surrounding environment, the grade of the bridge vibration can be used as its performance index, where the measurement of bridge vibration is the most effective function of bridge monitoring. Reinforcement and old processing are also carried out using this method.The monitoring system used this study is unique (bridge replacement method). In the future, when similar engineering methods are applied, these methods can also be improved based on the monitoring method adopted in this study to improve the quality and effect of monitoring.

## Figures and Tables

**Figure 1 sensors-19-05293-f001:**
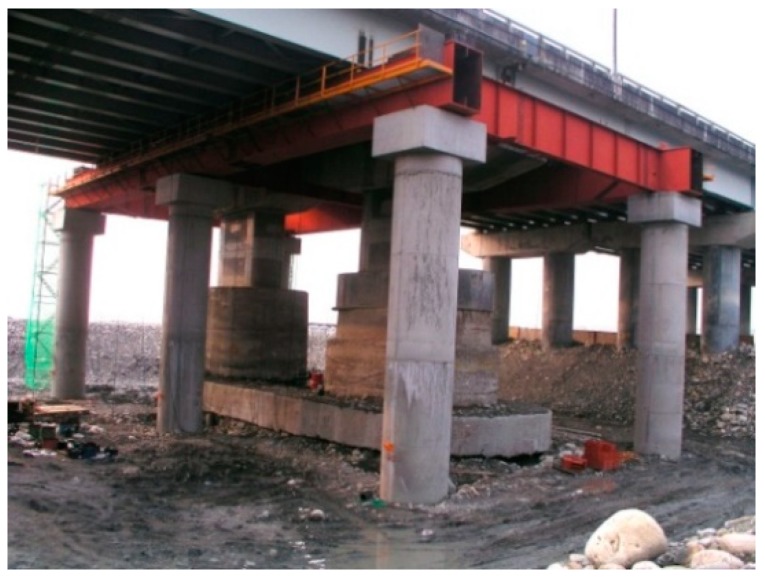
Temporary support steel frame.

**Figure 2 sensors-19-05293-f002:**
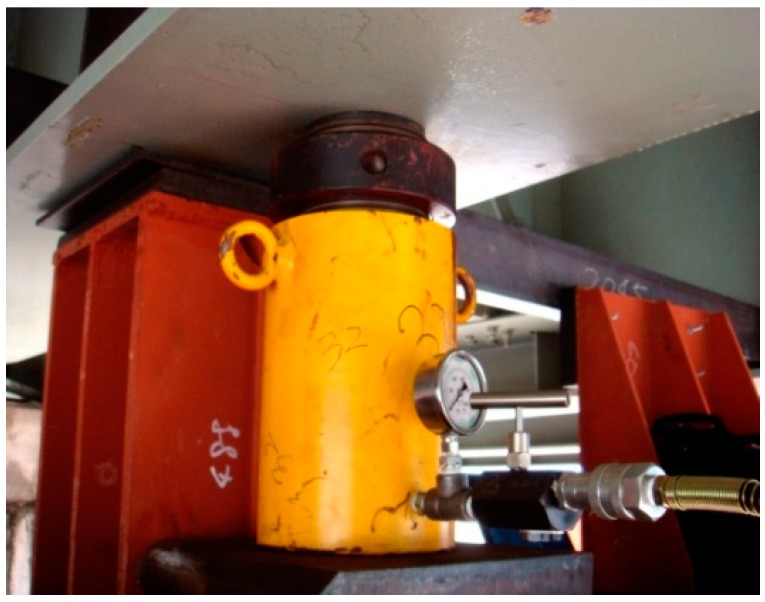
Jacking jack.

**Figure 3 sensors-19-05293-f003:**
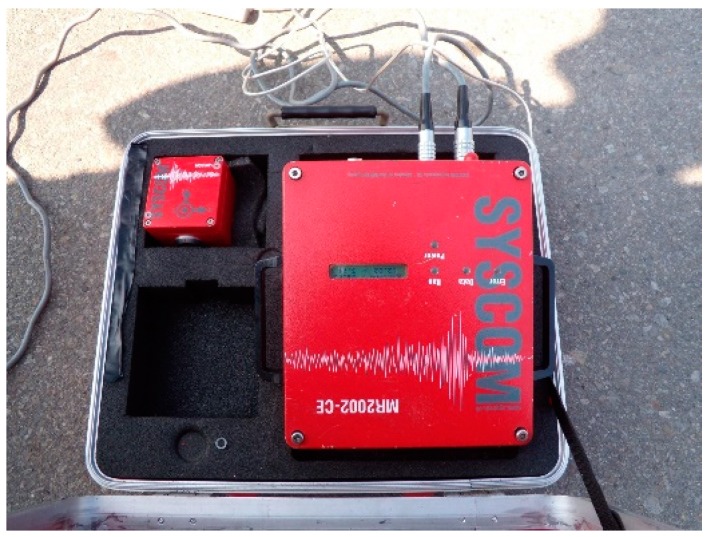
MR2002-CE type signal recorder.

**Figure 4 sensors-19-05293-f004:**
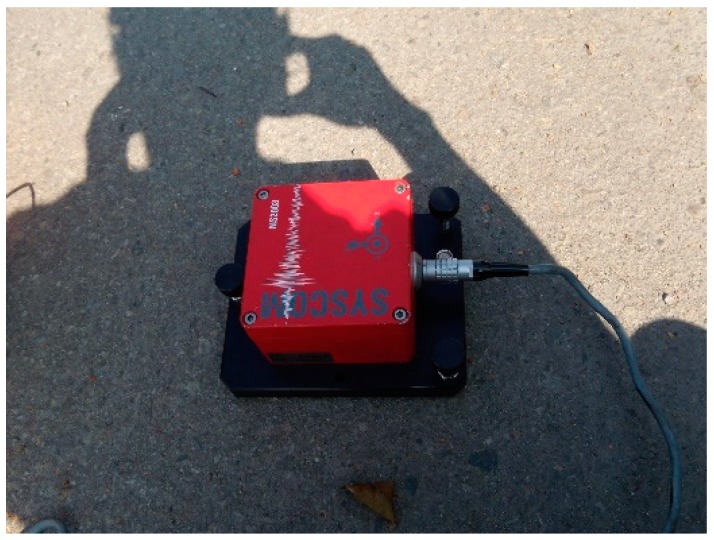
MS2003 speedometer.

**Figure 5 sensors-19-05293-f005:**
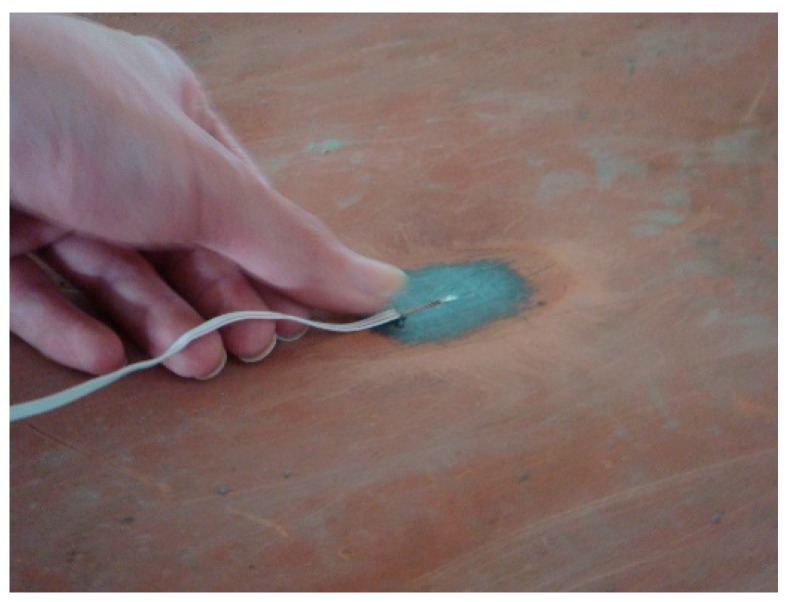
Adhesive strain gauge.

**Figure 6 sensors-19-05293-f006:**
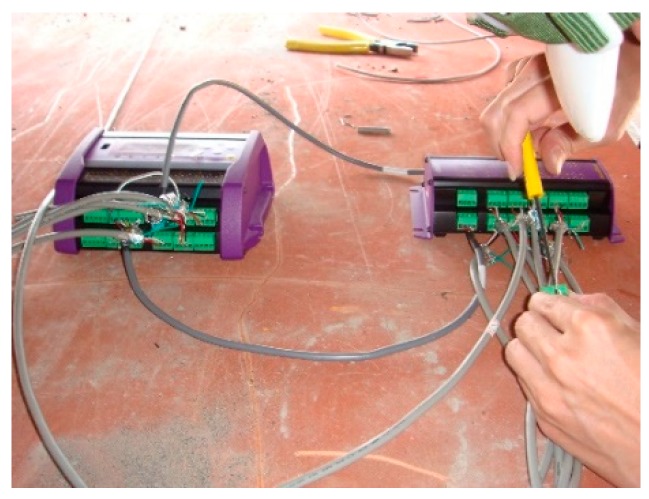
The strain gauge is connected to the data extractor.

**Figure 7 sensors-19-05293-f007:**
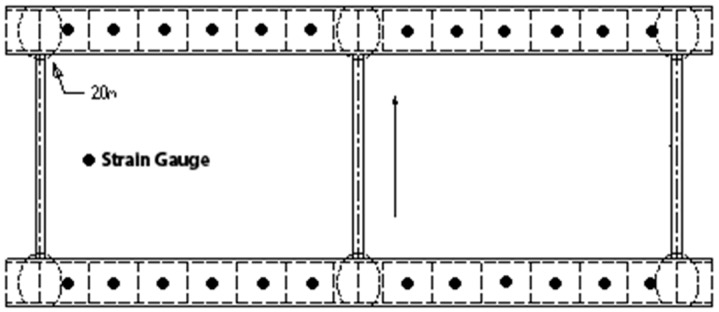
Schematic diagram of the position of the strain gauge (the black solid circle representative for the strain gauge’s positions).

**Figure 8 sensors-19-05293-f008:**
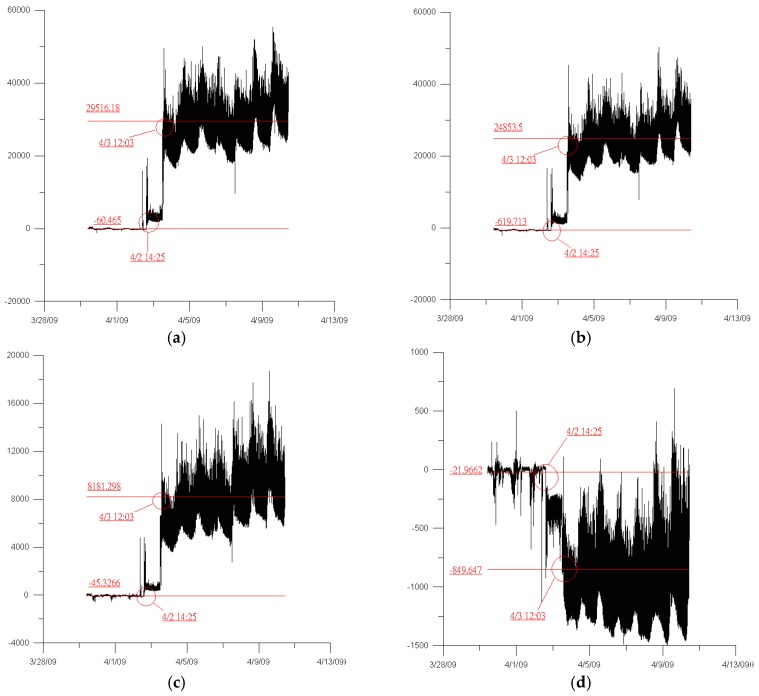
Changes from the strain gauge after jacking (**a**–**d**) shown the distribution of forces in different positions of strain gauges.

**Figure 9 sensors-19-05293-f009:**
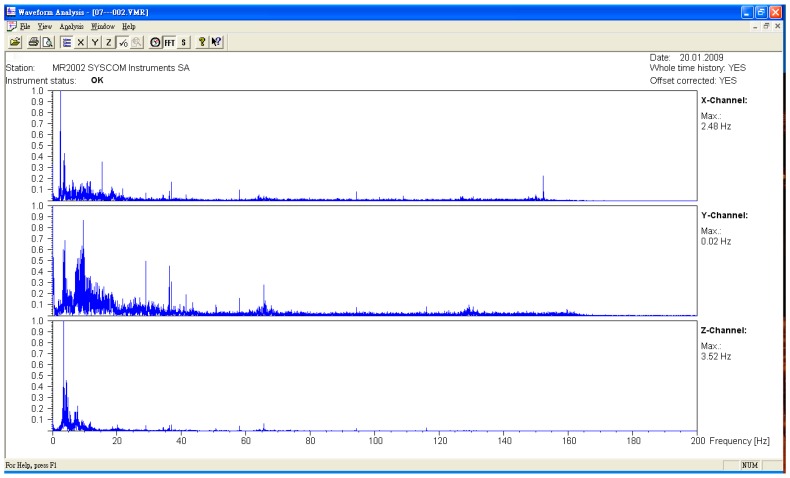
Original bridge frequency (P9).

**Figure 10 sensors-19-05293-f010:**
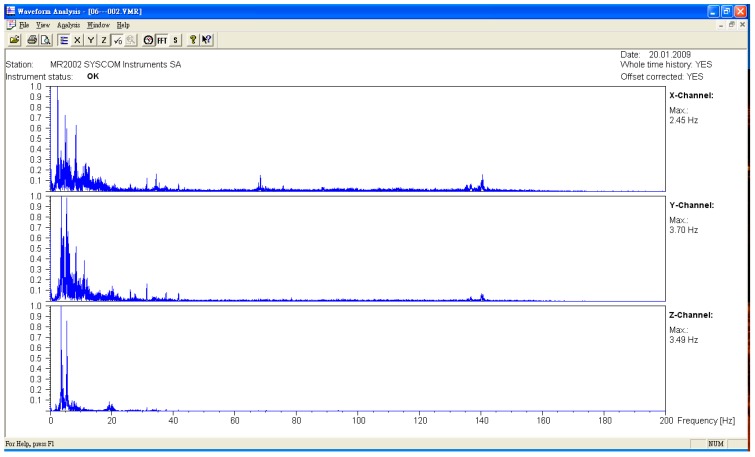
Original bridge frequency (P10).

**Figure 11 sensors-19-05293-f011:**
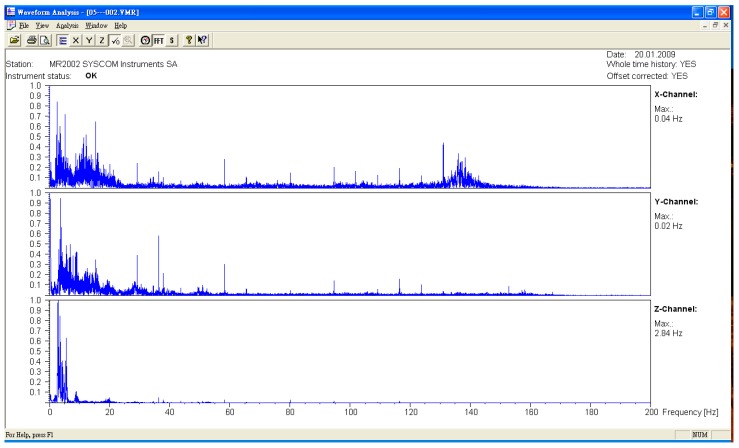
Original bridge frequency (P11).

**Figure 12 sensors-19-05293-f012:**
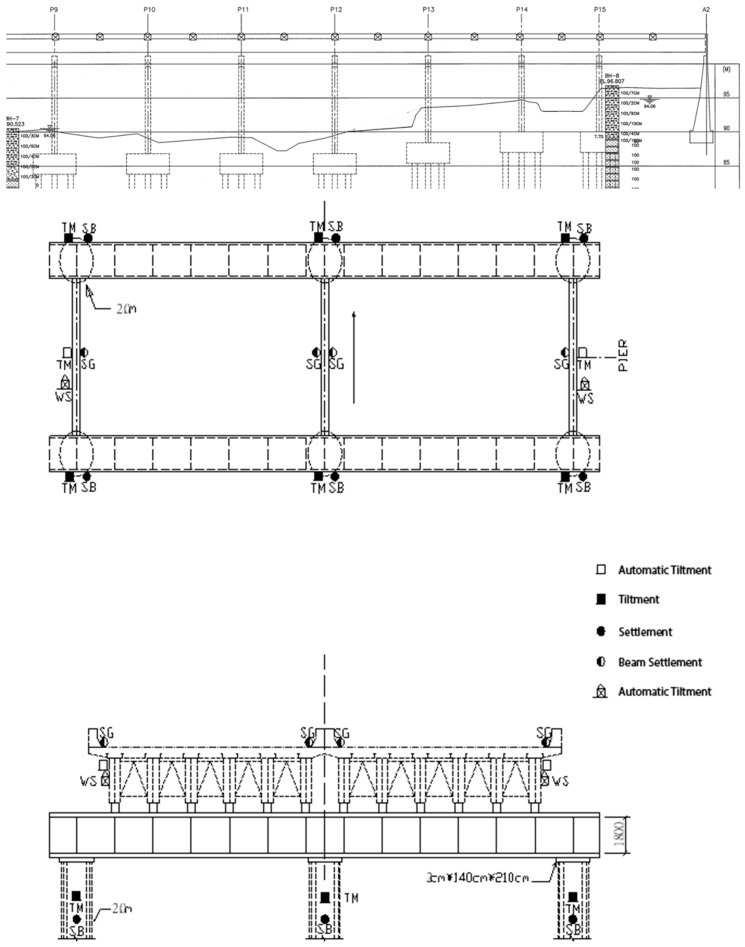
Configuration plan of each monitoring instrument (m).

**Figure 13 sensors-19-05293-f013:**
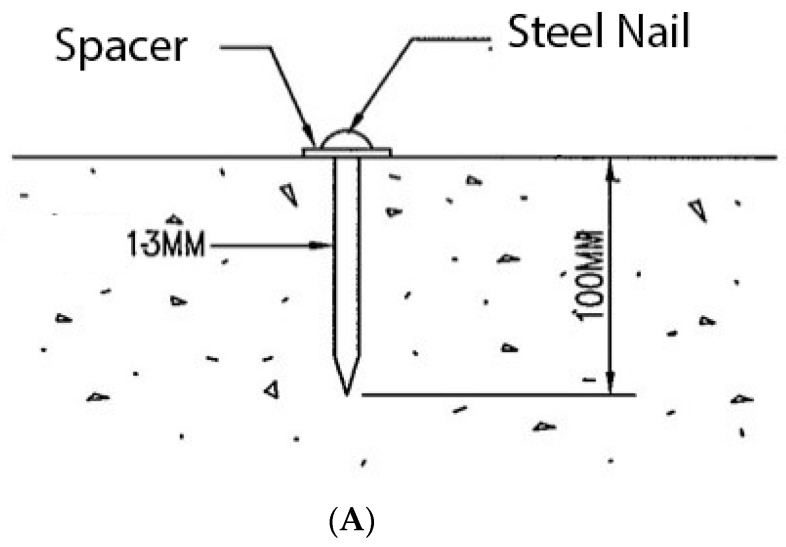
Schematic diagram of the sinking point installation. (**A**) Schematic diagram of the bridge deck test sink (SG) installation; (**B**) schematic diagram of the structure sinker point (SB) installation.

**Figure 14 sensors-19-05293-f014:**
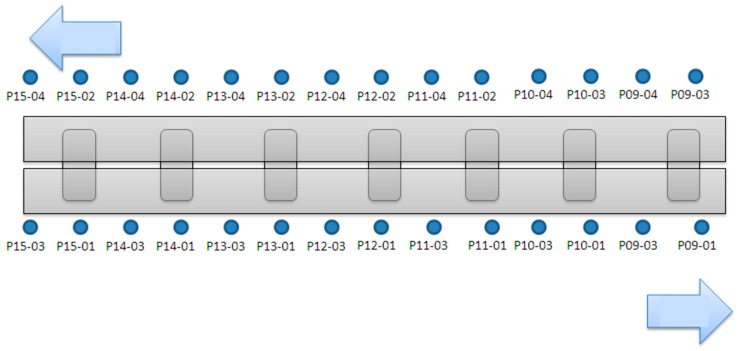
Plane layout diagram of the connected tube sinker.

**Figure 15 sensors-19-05293-f015:**
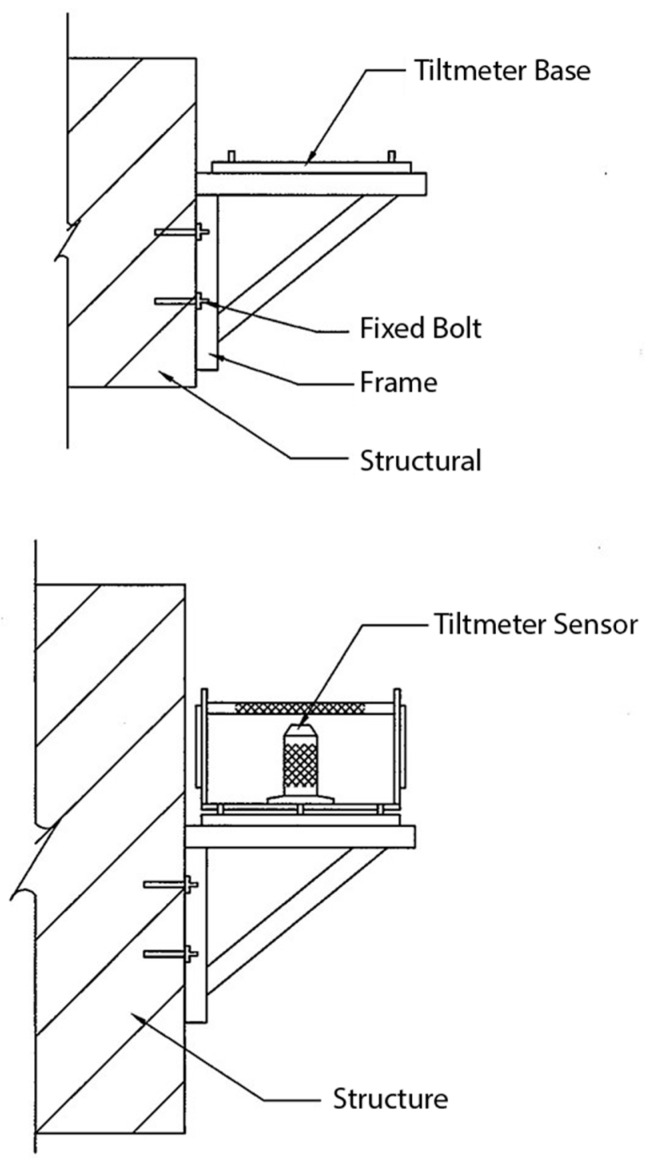
Tilt meter installation diagram.

**Figure 16 sensors-19-05293-f016:**
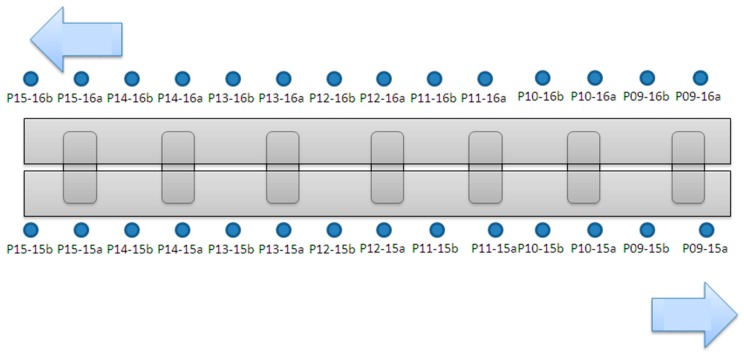
Electronic tilt meter plane configuration diagram.

**Figure 17 sensors-19-05293-f017:**
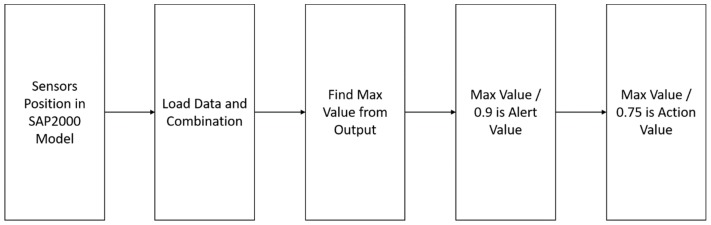
Flow chart of the management values of the static analysis, as in [6].

**Figure 18 sensors-19-05293-f018:**
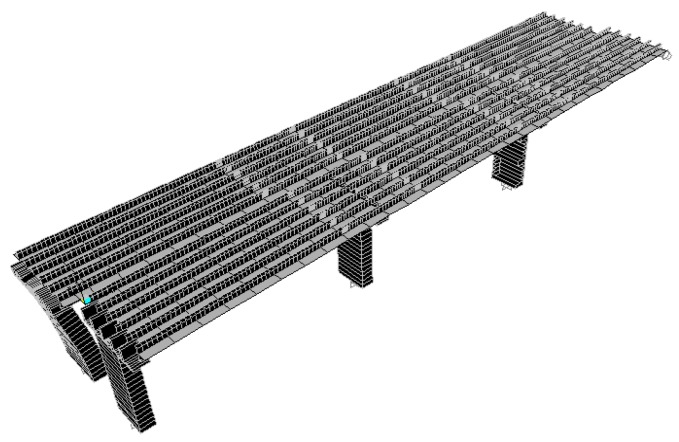
Original bridge model (1).

**Figure 19 sensors-19-05293-f019:**
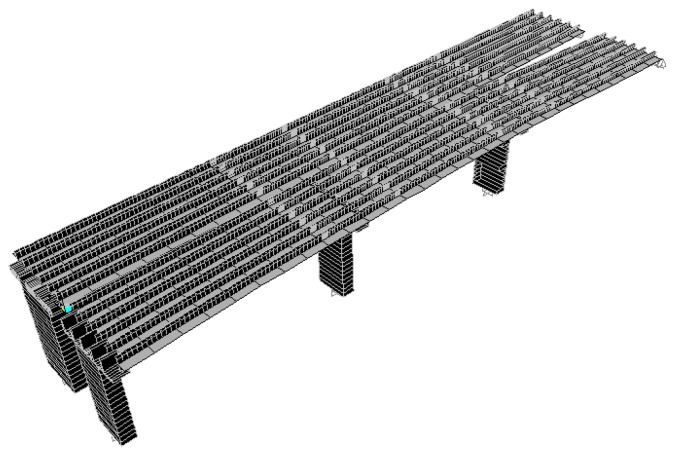
Original bridge model (2).

**Figure 20 sensors-19-05293-f020:**
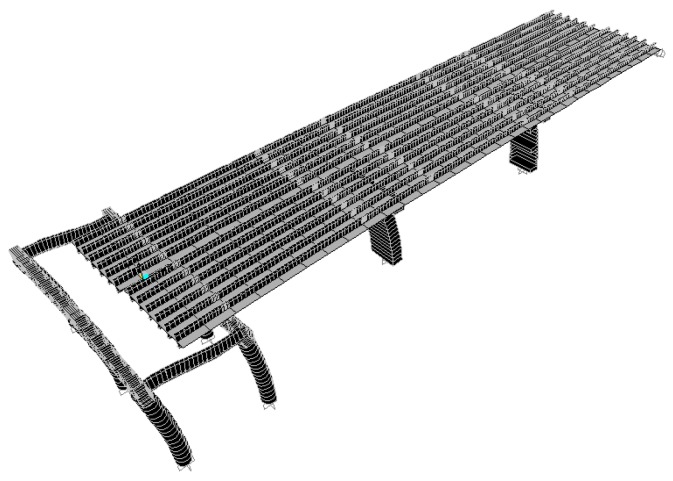
Bridge model after completion of P9 jacking (1).

**Figure 21 sensors-19-05293-f021:**
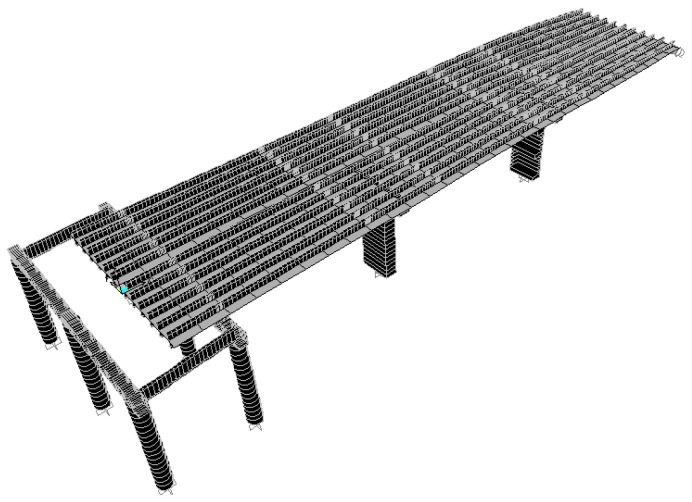
Bridge model after completion of P9 lift (2).

**Figure 22 sensors-19-05293-f022:**
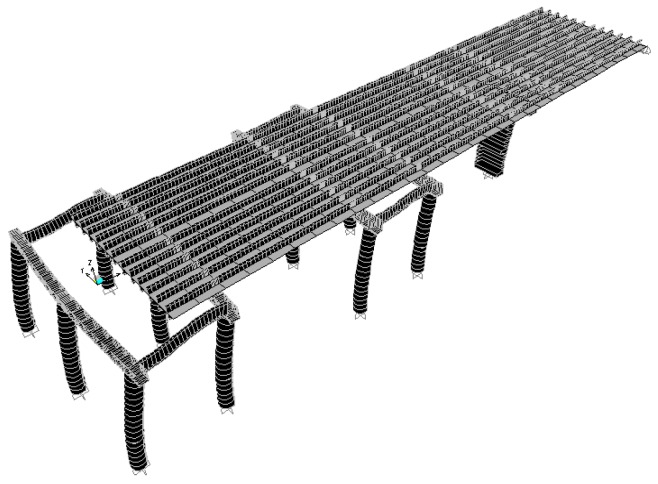
Bridge model after completion of P9 and P10 lifts (1).

**Figure 23 sensors-19-05293-f023:**
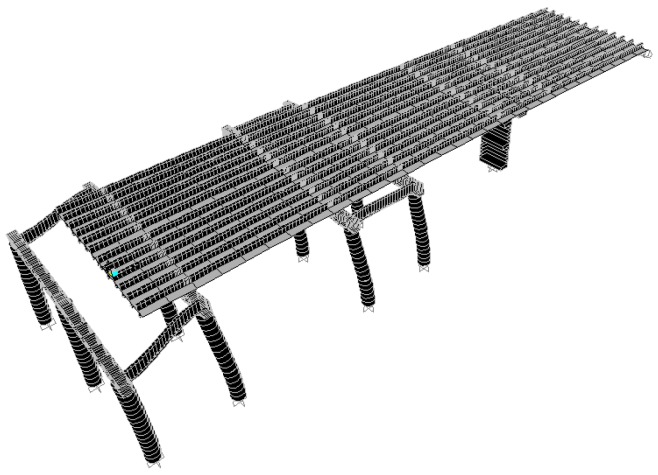
Bridge model after completion of P9 and P10 lifts (2).

**Figure 24 sensors-19-05293-f024:**
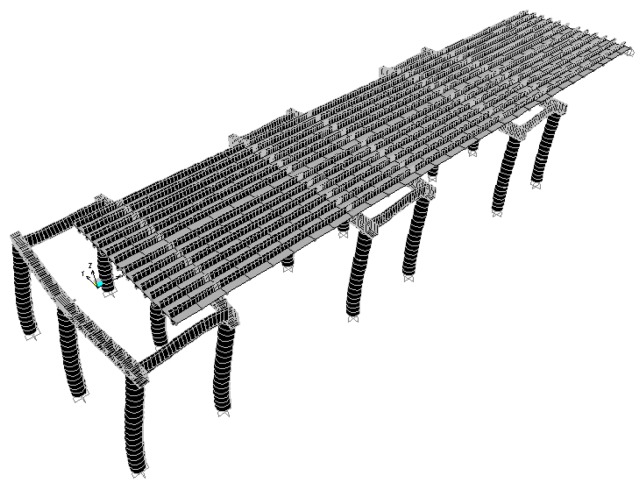
Bridge model after completion of P9, P10, and P11 lifts (1).

**Figure 25 sensors-19-05293-f025:**
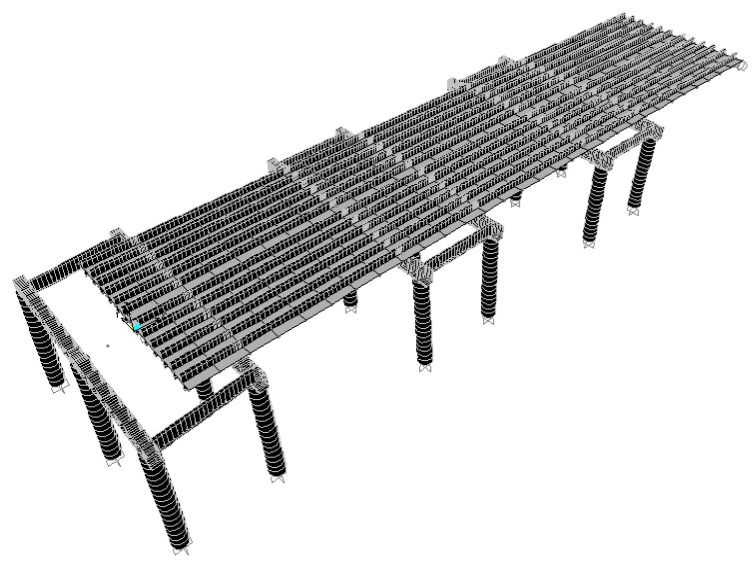
Bridge model after completion of P9, P10, and P11 lifts (2).

**Figure 26 sensors-19-05293-f026:**
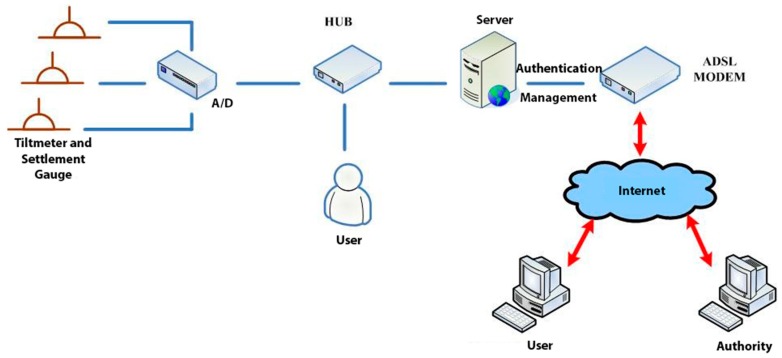
Software and hardware architecture diagram.

**Figure 27 sensors-19-05293-f027:**
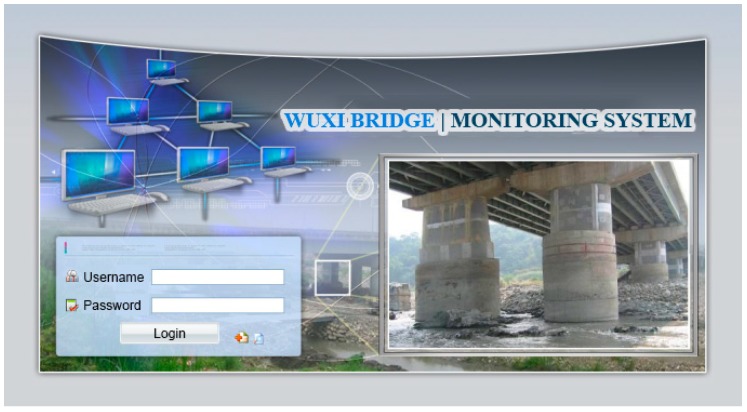
Login interface.

**Figure 28 sensors-19-05293-f028:**
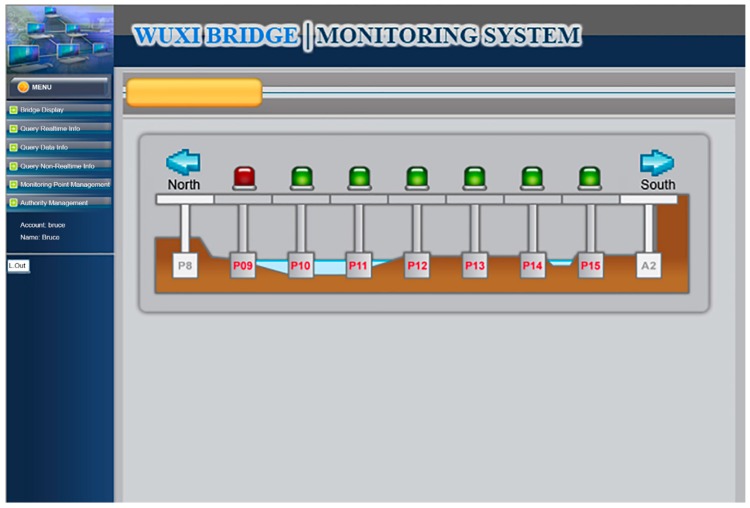
System home.

**Figure 29 sensors-19-05293-f029:**
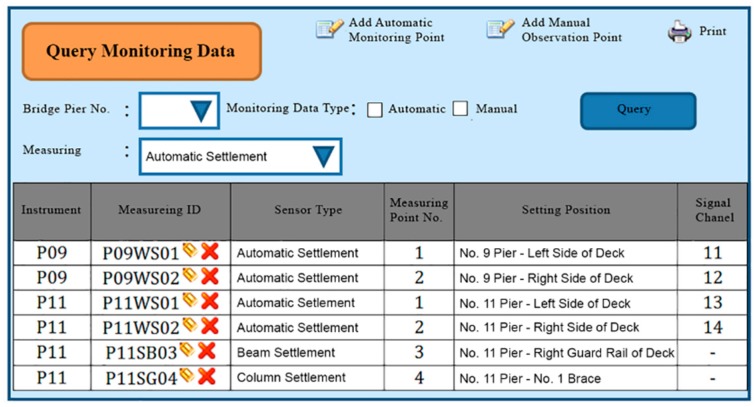
Monitoring point data query interface planning.

**Figure 30 sensors-19-05293-f030:**
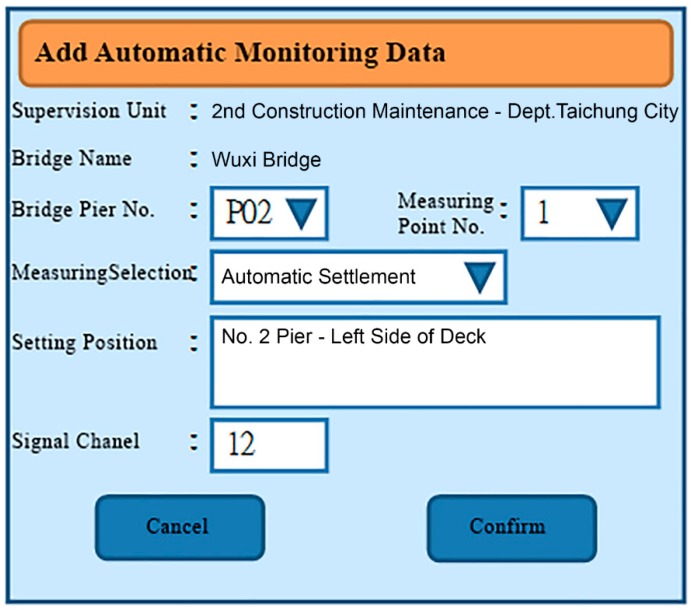
Automatic monitoring point information new interface planning.

**Figure 31 sensors-19-05293-f031:**
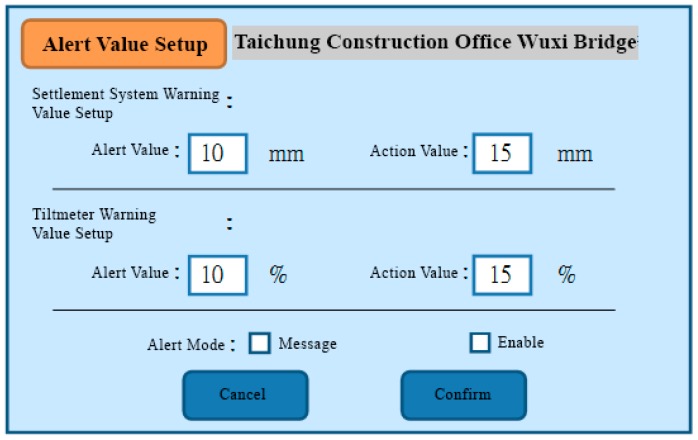
Warning value setting interface planning.

**Figure 32 sensors-19-05293-f032:**
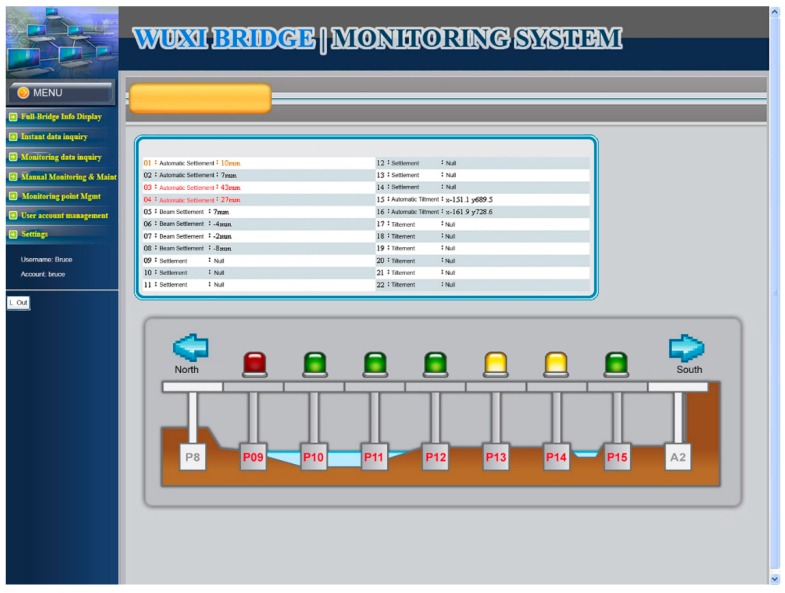
Full bridge monitoring interface planning.

**Figure 33 sensors-19-05293-f033:**
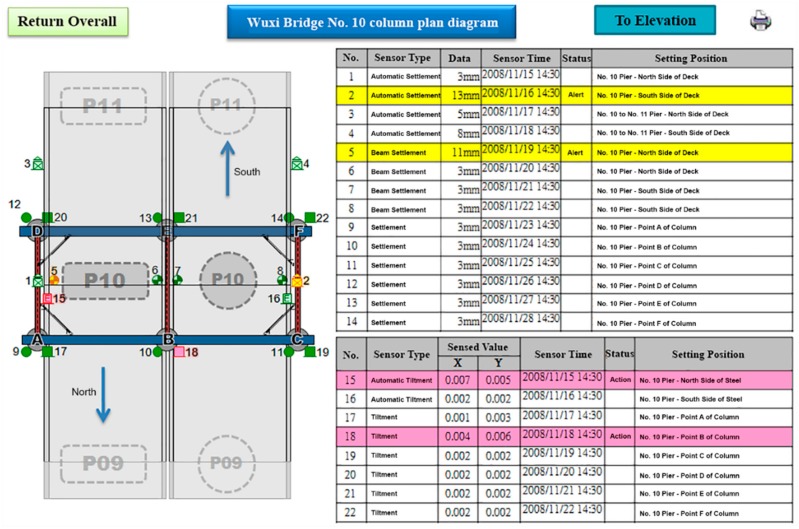
Bridge pier monitoring interface planning (Letter P09, P10, P11 identified for strain gauges’ position; circles shown the South direction; squares shown the North direction).

**Table 1 sensors-19-05293-t001:** Comparison of the bridge frequency before and after jacking.

Direction	*X* Axis		*Y* Axis		*Z* Axis	
Driving Direction	Northbound	Southbound	Northbound	Southbound	Northbound	Southbound
Unit	Frequency (Hz)	Frequency (Hz)	Frequency (Hz)	Frequency (Hz)	Frequency (Hz)	Frequency (Hz)
	P9					
Before jacking	2.48	2.48	3.67	3.58	3.52	3.48
After jacking	2.53	2.48	2.87	2.81	6.08	6.34
	P10					
Before jacking	2.45	2.43	3.41	3.65	3.47	3.47
After jacking	2.47	2.38	2.25	2.31	6.14	6.29
	P11					
Before jacking	2.49	2.48	3.65	3.67	3.65	3.47
After jacking	2.48	2.31	3.59	1.88	5.07	5.35

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
