# Peer review of "Automatic Management and Monitoring of Bridge Lifting: A Method of Changing Engineering in Real-Time"

_sensors, 2019, doi:10.3390/s19235293_

Round 1

Reviewer 1 Report

This paper introduces a bridge health monitoring system and its application to a real-life bridge structure, including the utilization and arrangement of various types of sensors. In addition to the length of this article, the paper is well-written and consistent with the scope of this journal. In the reviewer’s opinion, this paper can be considered for publication in this journal if the following comments are well addressed.

In lines 266-267, the finite element model is obtained by utilizing the Sap2000 software, and the authors mentioned that the “This field experiment data would be used to modify the vibration frequency of the SAP2000 bridge structure model, so that the model established by the SAP2000 software was more in line with the current analyzed situation.” This means that the finite element model updating procedure is employed to refine the FE model, which is a vitally important issue for the successful application of a structural health monitoring system. Although the authors focus on the application of an SHM system to a real-life bridge, they are still suggested to talk more about the information about the FE model updating algorithm and parameters selected for updating, etc., used in this software, which, however, is more important than solely introduce the application of the monitoring system. Another problem of this article is that the merit of this SHM system is not addressed as compared to the exsiting similar SHM system, which is suggested to be provided in the revised manuscript. In addition, the selection of threshold value for the alarm is not very clear according to the statement made in this original manuscript. In fact, this is a critical issue for all SHM systems when applied to the field application. The authors are suggested to provide some useful comments for this in the revised version.

Author Response

Response to Reviewer 1 Comments

This paper introduces a bridge health monitoring system and its application to a real-life bridge structure, including the utilization and arrangement of various types of sensors. In addition to the length of this article, the paper is well-written and consistent with the scope of this journal. In the reviewer’s opinion, this paper can be considered for publication in this journal if the following comments are well addressed.

Point 1: In lines 266-267, the finite element model is obtained by utilizing the Sap2000 software, and the authors mentioned that the “This field experiment data would be used to modify the vibration frequency of the SAP2000 bridge structure model, so that the model established by the SAP2000 software was more in line with the current analyzed situation.” This means that the finite element model updating procedure is employed to refine the FE model, which is a vitally important issue for the successful application of a structural health monitoring system. 

Response 1:

Dear the Editor and Reviewer 1:

Lifting of bridge will change bridge vibration mode and frequency. We can use microseismic sensor to measure all the frequencies of mode after lifting the bridge.

Point 2: Although the authors focus on the application of an SHM system to a real-life bridge, they are still suggested to talk more about the information about the FE model updating algorithm and parameters selected for updating, etc., used in this software, which, however, is more important than solely introduce the application of the monitoring system.

Response 2: Dear the Editor and Reviewer 1: …

SAP2000 is general-purpose civil-engineering software ideal for the analysis and design of any type of structural system. Basic and advanced systems, ranging from 2D to 3D, of simple geometry to complex, may be modeled, analyzed, designed, and optimized using a practical and intuitive object-based modeling environment that simplifies and streamlines the engineering process SAP2000 drives a sophisticated finite-element analysis procedure. An additional suite of advanced analysis features are available to users engaging state-of-the-art practice with nonlinear and dynamic consideration. Created by engineers for effective engineering, SAP2000 is the ideal software tool for users of any experience level, designing any structural system.

Point 3: Another problem of this article is that the merit of this SHM system is not addressed as compared to the exsiting similar SHM system, which is suggested to be provided in the revised manuscript. In addition, the selection of threshold value for the alarm is not very clear according to the statement made in this original manuscript. In fact, this is a critical issue for all SHM systems when applied to the field application. The authors are suggested to provide some useful comments for this in the revised version. 

Response 3: Dear the Editor and Reviewer 1: …

Base on the static and dynamic analysis result of the deformation, the monitoring and management value of each monitoring instrument were determined. According to the bridge design code, the bridge will bear the load data and combination. We find max value of deformation is safety value, max value divided by 0.9 is alert value and max value divided by 0.75 is action value.

Reviewer 2 Report

This paper shows an interesting monitoring work carried out in a bridge. However, this document seems to be more a technical report than a research paper. This monitoring work uses well-known sensors and all the process shows low novelty.

The introduction is too long and it is not well centered on the issue. The novelty and interest of this work is not highlighted enough.

Regarding the microseismic measurement experiment, I cannot find how the microseisms have been applied to the bridge.

Figures 9 to 11 have a poor presentation. Figure 12 has very poor quality. Figures 14 and 16 are not clear.

Figures 18 and 19 seem to be equal and they don't provide information. There are many figures about bridge models (figures 20 to 25). It is necessary to re-think about their usefulness or, at least, compact them into a unique figure.

This paper include many obvious paragraphs, explaining basic things and they should be omitted.

It is not well explained how the data provided from the in-field monitoring have been used in the SAP2000 FE model.

Some standards (like "Highway Bridge Design Code", among others) are not included in the References.

The explanation about the monitoring software is too long (9 pages). You should drastically reduce it.

Reference 25 is empty.

Author Response

Response to Reviewer 2 Comments

This paper shows an interesting monitoring work carried out in a bridge. However, this document seems to be more a technical report than a research paper. This monitoring work uses well-known sensors and all the process shows low novelty.

Point 1: The introduction is too long and it is not well centered on the issue. The novelty and interest of this work is not highlighted enough.

Response 1: Dear the Editor and Reviewer 2:

We did delete some common phrase words, and highlighted the newest point of this research (introduction part). We also cited more references (yellow highlighted 32-36, 37).

Point 2: Regarding the microseismic measurement experiment, I cannot find how the microseisms have been applied to the bridge.

Response 2: Dear the Editor and Reviewer 2:

We use microseismic measurement experiment included the environment test and the test of bridges. The environment ambient vibration test was firstly carried out before the bridge was constructed, and then was proceed after each ambient vibration test. Its goal are to verify bridge frequency and to identify the bridge vibration mode and to modify the bridge model accordingly.

Point 3: Figures 9 to 11 have a poor presentation. Figure 12 has very poor quality. Figures 14 and 16 are not clear.

Response 3: Dear the Editor and Reviewer 2:

To identify the frequencies of the bridge, the signals collected in time domain was transformed to the frequency domain by using the Fast Fourier Transform (FFT), as shown typically in Figures 9 to 11. There are many peaks appeared in the Fourier spectrum diagram due to the disturbance and noise in field.

Point 4: Figures 18 and 19 seem to be equal and they don't provide information. There are many figures about bridge models (figures 20 to 25). It is necessary to re-think about their usefulness or, at least, compact them into a unique figure.

Response 4: Dear the Editor and Reviewer 2:

Figures 18 and 19 are model1 and model2 of original bridge. These are Z axis of Northbound and Southbound.

Point 5: This paper include many obvious paragraphs, explaining basic things and they should be omitted.

Response 5: Dear the Editor and Reviewer 2:

We did delete some paragraph contained those common words in the introduction part (as highlighted by yellow color)

Point 6: It is not well explained how the data provided from the in-field monitoring have been used in the SAP2000 FE model.

Response 6: Dear the Editor and Reviewer 2:

The SAP2000 computer software is a powerful full-window interface structure analysis software, capable of the establishment of basic three-dimensional geometric shapes, the cross-sectional properties of rods and thin shell elements, the mechanical properties of reinforced concrete, steel structures, nonlinear elements, or the new definition of material properties. We use design diagram of bridge to build bridge model by Sap2000. According to vibration mode shapes and vibration frequencies of in-filed test result to modify Sap2000 model.

Point 7: Some standards (like "Highway Bridge Design Code", among others) are not included in the References.

Response 7: Dear the Editor and Reviewer 2:

We have added “Highway Bridge Design Code” promulgated by the Ministry of Communications in January 1990: Reference 37.

Point 8: The explanation about the monitoring software is too long (9 pages). You should drastically reduce it.

Response 8: Dear the Editor and Reviewer 2: Yes, we did delete Figure 34.

Point 9: Reference 25 is empty.

Response 9: Dear the Editor and Reviewer 2: Yes, we did make up that fault by add reference 25 (highlighted by yellow color).

Reviewer 3 Report

 The article introduces a method to perform the reparation of the bridge foundation without requiring its full closure. In the opinion of this reviewer, the article only presents minor innovations that might justify its publication in this journal. There are some aspects that need to be addressed.

The introduction section must be rewritten in order to clarify the concepts the authors want to use to indicate the contribution of their work. If the authors want to justify the method used to perform the reparation in the bridge, they must analyze alternative methods to perform this task. Please clearly indicate what are the contributions of the work. Please indicate the procedure used to retrofit the FE-based model. Please include a table showing the measured natural frequencies and the ones obtained using the FE model. What is the sampling frequency used by the designed monitoring system? Perform an extensive proofreading of the manuscript. There are several repeated words.

Author Response

Response to Reviewer 3 Comments

Comments and Suggestions for Authors

 The article introduces a method to perform the reparation of the bridge foundation without requiring its full closure. In the opinion of this reviewer, the article only presents minor innovations that might justify its publication in this journal. There are some aspects that need to be addressed.

Point 1: The introduction section must be rewritten in order to clarify the concepts the authors want to use to indicate the contribution of their work. If the authors want to justify the method used to perform the reparation in the bridge, they must analyze alternative methods to perform this task. Please clearly indicate what are the contributions of the work.

Response 1: Dear the Editor and Reviewer 3:

This paper defines a method to determine the monitoring system during reconstruction old bridge. According to on site test result and build bridge model, we can setup all the sensor thresholds. This is a real study and available to extend to another use case.

Point 2: Please indicate the procedure used to retrofit the FE-based model.

Response 2: Dear the Editor and Reviewer 3:

The SAP2000 computer software is a powerful full-window interface structure analysis software, capable of the establishment of basic three-dimensional geometric shapes, the cross-sectional properties of rods and thin shell elements, the mechanical properties of reinforced concrete, steel structures, nonlinear elements, or the new definition of material properties. We use design diagram of bridge to build bridge model by Sap2000. According to vibration mode shapes and vibration frequencies of in-filed test result to modify and retrofit bridge model.

Point 3: Please include a table showing the measured natural frequencies and the ones obtained using the FE model. (as same as the comments of Reviewer 2)

Response 3: Dear the Editor and Reviewer 3:

We use microseismic measurement experiment included the environment test and the test of bridges. The environment ambient vibration test was firstly carried out before the bridge was constructed, and then was proceed after each ambient vibration test. Its goal are to verify bridge frequency and to identify the bridge vibration mode and to modify the bridge model accordingly.

Point 4: What is the sampling frequency used by the designed monitoring system?

Response 4: Dear the Editor and Reviewer 3:

All the automatic sensor data sampling is per minute.

Point 5: Perform an extensive proofreading of the manuscript.

Response 5: Dear the Editor and Reviewer 3:

Yes, we have deleted. Thank you so much.

Round 2

Reviewer 1 Report

The reply to the Comment 2 of this Reviewer is not satisfied. The authors avoid to talk about the FE model updating procedure and the updating parameter selection process employed in this SHM system.

Author Response

Response to Reviewer 1 Comments

This paper introduces a bridge health monitoring system and its application to a real-life bridge structure, including the utilization and arrangement of various types of sensors. In addition to the length of this article, the paper is well-written and consistent with the scope of this journal. In the reviewer’s opinion, this paper can be considered for publication in this journal if the following comments are well addressed.

Point 1: In lines 266-267, the finite element model is obtained by utilizing the Sap2000 software, and the authors mentioned that the “This field experiment data would be used to modify the vibration frequency of the SAP2000 bridge structure model, so that the model established by the SAP2000 software was more in line with the current analyzed situation.” This means that the finite element model updating procedure is employed to refine the FE model, which is a vitally important issue for the successful application of a structural health monitoring system. 

Response 1:

Dear Editor and Reviewer 1:

Lifting of the bridge will change bridge vibration mode and frequency. We can use a microseismic sensor to measure all the frequencies of mode after lifting the bridge.

Point 2: Although the authors focus on the application of an SHM system to a real-life bridge, they are still suggested to talk more about the information about the FE model updating algorithm and parameters selected for updating, etc., used in this software, which, however, is more important than solely introduce the application of the monitoring system.

Response 2: Dear the Editor and Reviewer 1: …

The establishment of the FE bridge model can be divided into "whole bridge system" and "single partial system", which is the same as the vibration unit concept of the current bridge design code. The "whole bridge system" model is suitable for when the bridge type is geometrically irregular, such as a cable bridge, a horizontal multi-channel expansion joint, and the bridge is located in soft soil. The "single partial system" is suitable for quantifying the strength and stiffness capacity of a single frame, such as piers and is analyzed vertically and horizontally. The longitudinal model should consider the adjacent-span effect, and depends on the bridge length and the structure to decide whether to join the analysis model. The transverse model also considers the adjacent-span effect. The upper structure can be regarded as a rigid member. As for the abutment stiffness should be predicted and the analysis model should be added. In addition, the overall bridge analysis model must accurately describe the dimensions of all possible components and components, such as structural elements, thin-shell elements, springs, bearings and expansion joints, and other elements. Material properties, the behavior of the intact reaction structure. The section of the upper structure of the bridge is calculated for its section properties. According to the results of on-site microseismic measurement experiment of reinforced concrete bridges, the upper structure is a pre-force concrete. Torsional stiffness parameter is calculated in 200% of the full-section. The flexural rigidity of the horizontal axis parameter is calculated in 120% to 140% of the full-section. The flexural rigidity to the vertical axis parameter is calculated in 100% to 120% of the full-section. The cross-sectional of the rigid element is magnified 1000 times by the cross-sectional of the beam-column element.

Point 3: Another problem of this article is that the merit of this SHM system is not addressed as compared to the existing similar SHM system, which is suggested to be provided in the revised manuscript. In addition, the selection of threshold value for the alarm is not very clear according to the statement made in this original manuscript. In fact, this is a critical issue for all SHM systems when applied to the field application. The authors are suggested to provide some useful comments for this in the revised version. 

Response 3: Dear the Editor and Reviewer 1: …

Base on the static and dynamic analysis result of the deformation, the monitoring and management values of each monitoring instrument were determined. According to the bridge design code, the bridge will bear the load data and combination. We find max value of deformation is safety value, max value divided by 0.9 is alert value and max value divided by 0.75 is action value.

Submission Date

04 October 2019

Date of this review

29 Oct 2019 02:42:18

Reviewer 2 Report

Most of my comments have not been answered; only a few of them.

Author Response

Response to Reviewer 2 Comments

This paper shows an interesting monitoring work carried out in a bridge. However, this document seems to be more of a technical report than a research paper. This monitoring work uses well-known sensors and all the process shows low novelty.

Point 1: The introduction is too long and it is not well centered on the issue. The novelty and interest of this work are not highlighted enough.

Response 1: Dear the Editor and Reviewer 1:

We did delete some common phrase words and highlighted the newest point of this research (introduction part). We also cited more references (yellow highlighted 32-36, 37).

Point 2: Regarding the microseismic measurement experiment, I cannot find how the microseisms have been applied to the bridge.

Response 2: Dear the Editor and Reviewer 2:

We use microseismic to measure the environment test and the test of bridges. The environment ambient vibration test was firstly carried out before the bridge was constructed and then was proceed after each ambient vibration test. It measures the vibration of the bridge in three directions x, y, and z. The goal is to verify bridge vibration frequencies and to identify the bridge vibration modes and to modify the bridge model accordingly.

Point 3: Figures 9 to 11 have a poor presentation. Figure 12 has very poor quality. Figures 14 and 16 are not clear.

Response 3: Dear the Editor and Reviewer 2:

To identify the frequencies of the bridge, the signals collected in time domain was transformed to the frequency domain by using the Fast Fourier Transform (FFT), as shown typically in Figures 9 to 11. There are many peaks that appeared in the Fourier spectrum diagram due to the disturbance and noise in the field. There is 28 tube sinker on the Northbound and Southbound bridges as shown in Figures 14. There are 28 electronic tiltmeters on the Northbound and Southbound bridges as shown in Figures 16.

Point 4: Figures 18 and 19 seem to be equal and they don't provide information. There are many figures about bridge models (figures 20 to 25). It is necessary to re-think about their usefulness or, at least, compact them into a unique figure.

Response 4: Dear the Editor and Reviewer 2:

Figures 18 and 19 are model1 and model2 of the original bridge. These are Z-axis of Northbound and Southbound. After the bridge jacking and lift, the vibration mode of the bridge will be changed. The direction of the different vibrations can be seen from the figure20 and 21. Different bridge pillars will also cause different modes, as shown in the figure22 to 25.

There is different between Mode 1 and Mode 2:

Point 5: This paper includes many obvious paragraphs, explaining basic things and they should be omitted.

Response 5: Dear the Editor and Reviewer 1:

We did delete some paragraph contained those common words in the introduction part (as highlighted by yellow color)

Point 6: It is not well explained how the data provided from the in-field monitoring have been used in the SAP2000 FE model.

Response 6: Dear the Editor and Reviewer 2:

The establishment of the FE bridge model can be divided into "whole bridge system" and "single partial system", which is the same as the vibration unit concept of the current bridge design code. The "whole bridge system" model is suitable for when the bridge type is geometrically irregular, such as a cable bridge, a horizontal multi-channel expansion joint, and the bridge is located in soft soil. The "single partial system" is suitable for quantifying the strength and stiffness capacity of a single frame, such as piers and is analyzed vertically and horizontally. The longitudinal model should consider the adjacent-span effect, and depends on the bridge length and the structure to decide whether to join the analysis model. The transverse model also considers the adjacent-span effect. The upper structure can be regarded as a rigid member. As for the abutment stiffness should be predicted and the analysis model should be added. In addition, the overall bridge analysis model must accurately describe the dimensions of all possible components and components, such as structural elements, thin-shell elements, springs, bearings and expansion joints, and other elements. Material properties, the behavior of the intact reaction structure. The section of the upper structure of the bridge is calculated for its section properties. According to the results of on-site microseismic measurement experiment of reinforced concrete bridges, the upper structure is a pre-force concrete. Torsional stiffness parameter is calculated in 200% of the full-section. The flexural rigidity of the horizontal axis parameter is calculated in 120% to 140% of the full-section. The flexural rigidity to the vertical axis parameter is calculated in 100% to 120% of the full-section. The cross-sectional of the rigid element is magnified 1000 times by the cross-sectional of the beam-column element.

Point 7: Some standards (like "Highway Bridge Design Code", among others) are not included in the References.

Response 7: Dear the Editor and Reviewer 2:

We have added “Highway Bridge Design Code” promulgated by the Ministry of Communications in January 1990: Reference 37.

Point 8: The explanation about the monitoring software is too long (9 pages). You should drastically reduce it.

Response 8: Dear the Editor and Reviewer 1: Yes, we did reduce some contents and delete Figure 34.

Point 9: Reference 25 is empty.

Response 9: Dear the Editor and Reviewer 1: Yes, we did make up that fault by add reference 25 (highlighted by yellow color in the References).

Submission Date

04 October 2019

Date of this review

29 Oct 2019 02:42:18

Reviewer 3 Report

The authors have answered all the questions made by this reviewer.

Author Response

Point: The authors have answered all the questions made by this reviewer.

Response to Reviewer 3: Dear the Editor and Reviewer: We are sincerely thank you so much for your encouragement and supporting. Much appreciate!
